# Trends in health expectancies: a systematic review of international evidence

Gemma F Spiers ![ORCID], Tafadzwa Patience Kunonga ![ORCID], Fiona Beyer, Dawn Craig, Barbara Hanratty ![ORCID], Carol Jagger

Population & Health Sciences Institute, Newcastle University, Newcastle upon Tyne, UK

**Correspondence to**
Dr Gemma F Spiers;
gemma-frances.spiers@newcastle.ac.uk

## ABSTRACT

**Objectives** A clear understanding of whether increases in longevity are spent in good health is necessary to support ageing, health and care-related policy.

**Design** We conducted a systematic review to update and summarise evidence on trends in health expectancies, in Organisation for Economic Co-operation and Development (OECD) high-income countries.

**Data sources** Four electronic databases (MEDLINE, 1946–19 September 2019; Embase 1980–2019 week 38; Scopus 1966–22 September 2019, Health Management Information Consortium, 1979–September 2019), and the UK Office for National Statistics website (November 2019).

**Eligibility criteria** English language studies published from 2016 that reported trends in healthy, active and/or disability-free life expectancy in an OECD high-income country.

**Data extraction and synthesis** Records were screened independently by two researchers. Study quality was assessed using published criteria designed to identify sources of bias in studies reporting trends, and evidence summarised by narrative synthesis.

**Findings** Twenty-eight publications from 11 countries were included, covering periods from 6 to 40 years, between 1970 and 2017. In most countries, gains in healthy and disability-free life expectancy do not match the growth in total life expectancy. Exceptions were demonstrated for women in Sweden, where there were greater gains in disability-free years than life expectancy. Gains in healthy and disability-free life expectancy were greater for men than women in most countries except the USA (age 85), Japan (birth), Korea (age 65) and Sweden (age 77).

**Conclusion** An expansion of disability in later life is evident in a number of high-income countries, with implications for the sustainability of health and care systems. The recent COVID-19 pandemic may also impact health expectancies in the longer term.

## Strengths and limitations of this study

► To our knowledge, this is the first systematic review using reproducible methods to synthesise evidence about health expectancies and report quantitative comparisons of life and health expectancies to differentiate compression or expansion of morbidity.

► We used analyses published from 2016 to focus on the most contemporary evidence in trends.

► The quality of evidence was judged using criteria designed to assess threats to the validity of trends over time. However, due to the absence of methodological detail reported, it was not possible to give a clear judgement of study quality and bias for 10 of 28 studies included in the review.

of providing accessible, high quality and sustainable long-term care.[2–4] The growth in life expectancy is a positive, but with this comes a responsibility to ensure people have the support they need as they age, and to facilitate ageing in place.[5]

In 2019, the World Health Organisation (WHO) renewed its commitment to support countries to achieve longer and healthier lives with the Decade of Healthy Ageing 2020–2030 strategy.[6] A critical part of achieving longevity is understanding whether longer lives are typified by more years spent in good health (compression of disability) or poor health (expansion of disability). This has important implications for the provision of health and care services to respond to the needs of people as they age. It is, therefore, crucial to keep abreast of trends, specifically how the growth in life expectancy is matched by a growth in years spent in good health. Metrics to assess this most commonly include *healthy life expectancy* and *disability-free life expectancy*. Both provide an estimate of life expectancy spent in good health, but differ slightly with respect to their measurement. Healthy life expectancy tends to rely on single item questions of self-reported health, and is thus

## BACKGROUND

Populations are ageing worldwide. Globally, the proportion of those aged 65 and over has increased by 9% in the last two decades, and is expected to grow by a further 16% by 2050.[1] This demographic shift will require societies to adapt. If longer lives are spent in poor health, governments face the challenge

subject to fluctuations as expectations of health change over time.[7] Disability-free life expectancy is often calculated from multiple items about activity limitations and/or dependencies,[8] and therefore does not bear the same limitations as that of healthy life expectancy.

Previous reviews have summarised trends in total, healthy and disability-free life expectancy, the most recently in 2016.[8 9] Typically, such evidence shows that while people are living longer, gains in life expectancy are not consistently matched by a growth in the number of years lived in good health and free of disability. Nevertheless, this is an evolving evidence base requiring ongoing scrutiny. Longitudinal datasets, which often form the bedrock of the analysis of these trends, are continually acquiring new data, while in the UK, the Office of National Statistics (ONS) updates and publishes trends yearly. Regardless, the pressing policy relevance of these issues globally warrants an updated overview of trends in health expectancies. Understanding these issues in a global context is equally important in order to identify, and learn from, those countries whose populations are living healthier for longer. To obtain an up to date understanding of population trends in life expectancy and healthy ageing, we undertook a systematic review to (a) synthesise evidence about trends in health expectancies in high-income countries and (b) assess whether such trends are keeping pace with total life expectancy.

## METHODS

The methods used for this systematic review are reported according to the Preferred Reporting Items for Systematic Reviews and Meta-Analyses checklist and guidance.[10]

### Search strategy

A search strategy was developed using the concepts [*life expectancy*] AND [*trends* OR *impacting factors*]. The search was designed in MEDLINE using thesaurus headings and title, abstract and keyword field terms, and these elements translated to other databases. Electronic searches were carried out in MEDLINE (OVID) (1946–19 September 2019), Embase (OVID) 1980–2019 week 38, Scopus (1966–22 September 2019) and Health Management Information Consortium (OVID) (1979–September 2019), in October 2019 (see online supplemental materials for the strategy applied to MEDLINE). As this systematic review updated previous reviews,[8 9] searches were limited to studies published from 2016. The ONS website was also hand searched (November 2019) for reports published from 2016.

### Review criteria

Review criteria are summarised in table 1. Studies were included if they examined trends (ie, more than one time point) in health expectancies. Eligible health expectancies were general healthy life expectancy, disability free life expectancy, active life expectancy, health-related quality adjusted life expectancy and health adjusted

**Table 1** Review criteria

| | |
|---|---|
| Population | Studies must examine, trends from birth, 65 years and 85 years. Studies reporting trends from other ages were also reviewed where evidence was available. Studies must examine these trends in whole populations. Studies reporting trends in population subgroups only (ie, only those with heart failure) were ineligible. |
| Exposure | As this review reports evidence on life expectancy, health expectancy trends, an exposure variable was not required. |
| Comparator | Not applicable. |
| Outcome(s) | Active life expectancy, healthy life expectancy, disability-free life expectancy, health-related quality adjusted life expectancy, health-adjusted life expectancy. Studies reporting *only* life expectancy trends were ineligible. Studies must examine changes in these outcomes over time (ie, include more than one time point). Studies that report projections/forecasts of these outcomes were also eligible. |
| Study design | Studies must use an observational design and be carried out in an OECD high-income country. The review focused on evidence from the UK with comparison to evidence from other OECD high-income countries where possible. Studies published from 2016 were eligible. ONS reports were excluded if they were not the latest release, or reported trends for a period contained within a more recent ONS publication using the same data. |

OECD, Organisation for Economic Co-operation and Development; ONS, Office of National Statistics.

life expectancy. We did not include healthy cognitive life expectancy, dementia free life expectancy or life expectancy with diseases. Studies must be conducted in an Organisation for Economic Co-operation and Development (OECD) high-income country.[11] OECD high-income countries were selected for this review for comparability. Studies reporting trends in life expectancy *only* were ineligible. Where studies also reported prevalence of disability, dependency or self-rated health alongside the above outcomes, these were reported for context. For ONS reports, the most up-to-date analyses were included, with reports excluded if they were superseded by a more recently published analysis of the same data.

### Study selection

Titles and abstracts of all records were screened for relevance. The full texts of publications selected as relevant were retrieved and assessed for inclusion against the review criteria. Both stages of screening were undertaken independently by two researchers, with disagreements resolved through consensus.

**Table 2** Quality assessment criteria

| Criteria | Parameters |
|---|---|
| Comparability of interview methods between time points | Good: Identical<br>Fair: Change in mode<br>Poor: Change in disability, functioning or health outcomes |
| Quality of outcome measure | Good: Detailed multiple item measure<br>Fair: Single item global measure<br>*No criteria for poor* |
| Uses more than two time points to assess trend* | Good: Uses more than two time points<br>Fair: Uses only two time points<br>*No criteria for poor* |
| % Response in repeated cross-sectional studies* | Good: >70% response rate and <10% drop in subsequent surveys<br>Fair: <70% or >10% drop in subsequent surveys<br>*No criteria for poor* |
| Loss to follow-up | Good: NA or <5%<br>Fair: 5%–10%<br>Poor: >10%<br>Note: This only applies to longitudinal study designs (ie, not independent repeat cross-sections) |
| Proportion of proxy interviews | Good: <10%<br>Fair: 10%–20%<br>Poor: >20% |
| Proportion of missing data | Good: <5%<br>Fair: 5%–10%<br>Poor: >10% |

*Added to the quality assessment following expert advice.

## Quality appraisal

Study quality and bias were assessed using an adapted version of previously published criteria for studies reporting trends.[12] This approach assesses 'threats to the validity of comparisons over time' (p3140). Following expert advice, we also judged two further criteria: whether the trend analysis used more than two time points, and the % of non-response in repeated cross-section surveys. Criteria were rated as *good*, *fair* or *poor*, according to the parameters summarised in table 2. Where study publications did not report the required information to assess quality, other associated publications (eg, methodological and technical reports for the datasets used) were consulted, and we attempted to contact all authors and/or administrators of the datasets used. If the required information was not available from these sources, the criterion was assessed as *unclear*.

Using the assessments (good, fair, poor or unclear) for each criterion, studies were given a summary rating of quality. As studies were often based on summaries of cross-sectional data, they did not typically report information to assess the criterion *proportion lost to follow-up*. We also struggled to find information to assess the criteria *proportion of proxy interviews* and *proportion of missing data*. We, therefore, based our summary judgement on four criteria: the comparability of methods over time, quality of the outcome measure, response rate or loss to follow-up, and whether more than two time points were used. Studies with all four criteria rated good received a summary judgement of good. Studies with one or more criteria rated poor received a summary judgement of poor. The remainder were rated fair.

Where studies used multiple outcomes and were given different judgements for each, this was noted in the summary rating.

### Data extraction and synthesis

Study details (author, publication date, country, study design) and trend estimates (for each time point measured, and the change between the first and last time point) were extracted onto an Excel spreadsheet by one researcher. Fifty per cent of studies were checked for accuracy by a second researcher. Where publications did not report the change between the first and last time point, we calculated this by subtracting the first time point estimate from the last time point estimate.

A narrative synthesis was used to summarise evidence on trends by outcome (eg, healthy life expectancy, disability-free life expectancy, disability or dependency prevalence), supported by data summary tables. Estimates of changes in each health expectancy were compared with changes in life expectancy. This comparison provides evidence of whether there had been an expansion (health expectancy gains are smaller than life expectancy gains) or compression (health expectancy gains are equivalent to or greater than life expectancy gains) of disability and ill health.

### Patient and public involvement

This review was requested by our funder within a timescale that did not allow for meaningful public and patient involvement.

## FINDINGS

Twenty-eight studies met the review criteria (figure 1, and online supplemental table 1). Seven studies reported trends in the UK,[7 8 13–17] including England,[7 8 14] England and Wales,[13 15 17] and each of the four devolved countries and the UK as a whole.[16] Two of the UK studies were ONS reports.[15 16] The remaining studies reported trends in Belgium,[17–19] Canada,[20] Denmark,[21] Japan,[22 23] the Netherlands,[24 25] Norway,[26] Republic of Korea,[27 28] Sweden,[29 30] Switzerland[31] and the USA.[32–35] Three Global Burden of Disease studies were included, which reported trends across multiple countries, and high-income countries combined.[36–38] For these three studies, we used data for all high-income countries combined.

The assessment of study quality is detailed in online supplemental table 2. Three studies were rated good,[20 34 35] eleven were rated fair,[8 13 14 18 19 21 22 24 30 32 33] one was rated good *and* fair (as it used two outcome measures that each received a different quality rating)[7] and three were rated poor.[26 29 31] Ten studies were rated unclear due to a lack

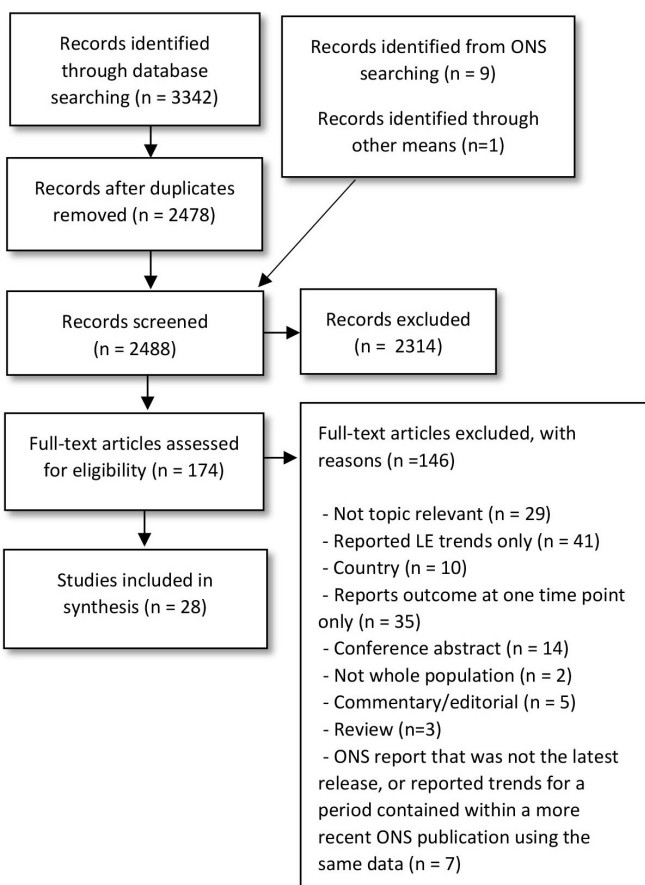

**Figure 1** Preferred Reporting Items for Systematic Reviews and Meta-Analyses flowchart.

of information required to assess quality on the four key criteria.[15–17 23 25 27 28 36–38] As majority large proportion of studies received a summary rating of unclear, the synthesis below does not prioritise evidence with a higher quality rating. Instead, the summary judgement is provided in online supplemental table 2 for the reader's reference.

Across most studies, trends demonstrated a growth in health expectancies, with gains typically greater for men than women (online supplemental tables 3a–d, 4a–f, 5a–d, 6a–c, 7a–b). Exceptions where changes were greater for women include the USA (age 85),[33] Japan (birth),[23] Korea (age 65)[28] and Sweden (age 77).[30] One analysis of UK data over the more recent period of 2009/2012 to 2015/2017 also indicated that healthy life expectancy declined for women by 0.2 years.[15] Disability-free life expectancy at age 20–64 years also declined by 0.6 years between 1970 and 2010 for women in the USA in one study.[33]

### Is there evidence of an expansion or compression of poor health and disability?

Table 3 summarises the change in each life expectancy and health expectancy across trend periods for each study where comparisons were possible. The comparison does not include studies where: the metric of change was not comparable between life expectancy and health expectancy[18 20]; trends were reported as a slope of index

inequality only[16]; only forecasts were reported[13 14 32] and life expectancy is reported only in a graph.[29 34 35]

There is evidence from observed (ie, not forecasted) trends in the UK that changes in health expectancies were smaller than those in life expectancy. This is consistent with the evidence of a reduction in the proportion of life spent without disability in the UK. This pattern was also observed in: all high-income countries combined in the Global Burden of Disease studies,[36–38] Norway,[26] Belgium,[19] Japan[22 23] and the USA[33] for all, and in Switzerland[31] and Sweden[30] for men but not women. This points to evidence of an expansion of disability in a number of high-income countries, although not always consistently between men and women.

Gains in disability-free life expectancy were greater than gains in life expectancy for women in Sweden, a finding that was evident in two different datasets.[30] This indicates a compression of disability for women in Sweden. In another study, gains in life expectancy and healthy life expectancy were similar for women in Switzerland,[31] which might also suggest a compression of disability. However, this finding should be interpreted with some caution: the quality assessment highlighted a potential threat to the validity of the trend due to changes in the phrasing of, and response items for, the outcome measure between data waves.

Evidence from Korea and the Netherlands was less certain. Contrasting findings between studies gave an unclear picture about whether these countries had experienced a compression or expansion of poor health. For example, in one study from Korea, changes in quality adjusted life expectancy at birth were greater than changes in life expectancy.[27] Another study, using the same data but a different measure (healthy life expectancy), reported that gains at birth still lagged behind those for life expectancy, although gains in healthy life expectancy at age 65 and 85 years were slightly higher than gains in life expectancy over the study period.[28] This contrast may reflect the difference in measures; Jo and colleagues suggest that quality adjusted life expectancy may be overestimated using the EQ-5D-3L, which forms the basis of this measure.[27]

Three studies from the Netherlands also offered inconsistent findings: this may reflect differences in the trend periods, age at which expectancies were estimated and the measures used.[17 24 25] For example, Reus-Pons and colleagues demonstrated gains in healthy life expectancy were smaller than gains in total life expectancy at age 50 years, over a 10-year period (2001–2011). Deeg and colleagues examined physical healthy life expectancy over a 23-year period (1993–2016): a decline was observed, while total life expectancy increased. Gheorghe and colleagues estimated quality adjusted life expectancy at age 25 and 65 years, and stratified by educational attainment. These trends indicated an expansion of poor health at age 25 and 65 years for men at all levels of educational attainment, but a compression of poor health for

**Table 3** Change in life expectancy, healthy life expectancy and disability-free life expectancy across all studies where reported*

| Study | Country | Age | Trend period | Change in LE | Change in HLE | Change in DFLE |
|---|---|---|---|---|---|---|
| All high-income countries combined from the Global Burden of Disease studies | | | | | | |
| GBD study 2016 | All high income | 0 | 2005, 2015 | Men: 1.75 Women: 1.33 | Men: 1.43 Women: 1.08 | – |
| GBD study 2017 | All high income | 0 | 1990, 2016 | Men: 5.63 Women: 4.14 | Men: 4.43 Women:3.21 | – |
| | | 65 | | Men: 3.51 Women: 3.03 | Men: 2.27 Women: 2.6 | – |
| GBD study 2018 | All high income | 0 | 1990, 2017 | Men: 5.6 Women: 4.2 | Men: 4.2 Women: 3.0 | – |
| Studies with samples from Europe | | | | | | |
| Jagger et al[7] | England | 65 | 1991, 2011 | Men: 4.5 Women: 3.6 | Men: 3.8 (3.5–4.1) Women: 3.1 (2.7–3.4) | Men: 2.6 (2.3–2.9) Women: 0.5 (0.2–0.9) |
| Kingston et al[8] | England | 65 | 1991, 2011 | Men: 4.7 Women: 4.1 | – | Men: 1.7 (1.2–2.1) Women: 0.2 (−0.4 to 0.7) |
| Reus-Pons et al[17] | England and Wales | 50 | 2001, 2011 | Men: 2.8 Women: 2.2 | Men: 0.25 Women: −0.15 | – |
| ONS[15] | UK | 0 | 2009/2011–2015/2017 | Men: 0.8 Women: 0.4 | Men: 0.4 Women: −0.2 | – |
| Bronnum-Hansen et al[21] | Denmark | 65 | 2006/2007, 2010/2011, 2013/2014 | Difference in change in LE between high and low education: Men: 0.1 Women: 0.3 | – | Difference in change in DFLE between high and low education Men: −0.3 Women: −0.3 |
| Deeg et al[24] | Netherlands | 65 | 1993, 1996, 1999, 2002, 2006, 2009, 2012, 2016 | Men: 4.0 Women: 2.2 | Physical Men: −2.2 Women: −1.5 | – |

Continued

**Table 3** Continued

| Study | Country | Age | Trend period | Change in LE | Change in HLE | Change in DFLE |
|---|---|---|---|---|---|---|
| Gheorghe et al[25] | Netherlands | 25 | 2001, 2011 | Men:<br>High: 2.88<br>Med: 3.22<br>Low: 2.43<br>Women:<br>High: 1.78<br>Med: 1.53<br>Low: 1.15 | Men:<br>High 2.85<br>Med: 3.01<br>Low: 2.13<br>Women:<br>High: 2.64<br>Med: 1.57<br>Low: 1.80 | – |
| | | 65 | | Men:<br>High: 2.48<br>Med: 2.24<br>Low: 1.68<br>Women:<br>High: 1.64<br>Med: 1.41<br>Low: 1.12 | Men:<br>High: 2.17<br>Med: 1.91<br>Low: 1.37<br>Women:<br>High: 1.74<br>Med: 1.21<br>Low: 1.14 | – |
| Reus-Pons et al[17] | Netherlands | 50 | 2001, 2011 | Men: 2.82<br>Women: 1.84 | Men: 2.21<br>Women: 1.25 | – |
| Remund et al[31] | Switzerland | 30 | 1990/1994, 1995/1999, 2004/2004, 2010/2014 | Men: 5.02<br>Women: 3.09 | Men: 4.52<br>Women: 3.09 | – |
| Storeng et al[26] | Norway | 50 | 1984/1986, 1995/1997, 2006/2008 | Men: 6.99 (5.27–8.72)<br>Women: 6.75 (5.16–8.34) | Men: 6.90 (6.08–7.73)<br>Women: 5.40 (4.56–6.25) | Men: 2.71 (2.01–3.42)<br>Women: 0.33 (−0.40 to 1.06) |
| Sundberg et al[30] | Sweden (SWEOLD) | 77 | 1992, 2002, 2004, 2011 | Men: 1.7<br>Women: 1.1 | – | Men: 1.1<br>Women: 1.6 |
| | Sweden (SHARE) | | 2004, 2011 | Men: 0.6<br>Women: 0.4 | – | Men: 0.1<br>Women: 1.3 |
| Yokota et al[29] | Belgium | 15 | 2001, 2004, 2008 | Men: 1.6<br>Women: 1.0 | – | Men: 0.7<br>Women: −0.7 |
| Studies with samples from Asia | | | | | | |
| Jo et al[27] | R. Korea | 0 | 2005, 2007, 2008, 2009, 2010, 2011, 2012, 2013 | Men: 3.38<br>Women: 3.15 | Men: 4.03<br>Women: 3.44 | – |
| Lee et al[28] | R. Korea | 0 | 2005, 2008, 2011 | Men: 2.5<br>Women: 2.6 | Men: 1.4<br>Women: 1.2 | – |
| | | 65 | | Men: 1.6<br>Women: 2.0 | Men: 2.2<br>Women: 2.6 | – |
| | | 85 | | Men: 0.3<br>Women: 0.7 | Men: 1.4<br>Women: 1.4 | – |

Continued

**Table 3** Continued

| Study | Country | Age | Trend period | Change in LE | Change in HLE | Change in DFLE |
|---|---|---|---|---|---|---|
| Sugawara et al[22] | Japan | 0 | 2000, 2010 | Men: 1.9<br>Women: 1.7 | – | Men: 1.0<br>Women: 0.4 |
| Tokudome et al[23] | Japan | 0 | 1990, 1995, 2000, 2005, 2010, 2013 | Men: 4.01<br>Women: 4.43 | Men: 3.02<br>Women: 3.32 | – |
| Studies with sample from North America | | | | | | |
| Crimmins et al[33] | USA | 0 | 1970, 1980, 1990, 2000, 2010 | Men: 9.2<br>Women: 6.4 | – | Men: 4.5<br>Women: 2.7 |
| | | 20–64 | | Men: 1.8<br>Women: 0.9 | – | Men: 0.9<br>Women: –0.6 |
| | | 65 | | Men: 4.7<br>Women: 3.5 | – | Men: 2.7<br>Women: 2.4 |
| | | 85 | | Men: 1.1<br>Women: 1.3 | – | Men: 0.5<br>Women: 0.8 |

*Table and comparison does not include studies where: the metric of change was not comparable for each health expectancy (Steensma et al[20]); trends were reported as slope of index inequality only (ONS[19]); only forecasts are reported (Kingston et al[14], Guzman-Castillo et al[13], Cao et al[32]); total life expectancy is reported only as a graph (Lagergren et al[29], Freedman and Spillman[34], Freedman et al[35]); trends are reported as difference in change in DFLE between levels of education and not comparable to the LE trend (Renard et al[18]).

DFLE, Disability-free life expectancy; GBD, Global Burden of Disease; HLE, Healthy life expectancy; LE, Life expectancy.

women at age 25 years (all levels of education) and 65 (high and low education).

## DISCUSSION

This systematic review was undertaken to update our current understanding of trends in health expectancies in OECD high-income countries. The principal finding is that changes in health expectancies have not kept pace with the growth in life expectancy in a number of high-income countries. One clear exception was Sweden, where gains in women's disability-free life expectancy were greater than gains in life expectancy over a period of almost 20 years. This was a finding evidenced from two different datasets within Sundberg and colleagues' study (SWEOLD: 1992–2011; SHARE: 2004–2011). This finding contrasts with an earlier review, which found evidence of a compression of disability for men, but not women, in Sweden.[9] This has now been reversed with an expansion of disability for men.[30] Sundberg and colleagues attribute the compression of disability for women to improved health for women but not men, while men were living longer due to falling deaths from cardiovascular disease.

The equivalence of total and healthy life expectancy for women in Switzerland also indicates a compression of poor health in Switzerland, although as noted earlier, this finding should be interpreted with some caution. Nevertheless, based on the countries with data, our findings suggest that countries still need to make significant progress to achieve the WHO's Decade of Healthy Ageing goal of healthier, longer lives for all.

The observed expansion of disability and poor health has important implications for older people's health and care. Across many high-income countries, care services that support older people's day to day independence and quality of life are not universal but subject to payment barriers.[3 4] Living longer with greater disability and in poor health signals a need for greater policy focus on ensuring older people have timely access to care. This is a particularly timely message for policy makers in the UK, as the All Party Parliamentary Group for Longevity set out recommendations to achieve the UK Government's goal of extending healthy life by 2035 and reducing inequalities.[39] Furthermore, the finding that women are living with longer periods of disability may warrant a particular focus on how services can support women in later life. This is especially important given that women are more likely than men to experience financial insecurity in later life,[40] and thus may face greater barriers to paid-for care services.

Finally, given the high death rates in some countries as a result of the COVID-19 pandemic, and the likely long-term health consequences of the virus that are becoming apparent, further scrutiny of future trends in health expectancies is needed. A recent analysis of UK mortality data in 2020, for example, indicates that COVID-19 has reduced average life expectancy by around a year.[41] The

ways in which this may impact health expectancies has yet to be determined.

### Strengths and limitations

This systematic review provides a robust picture of trends in health expectancies in high-income countries. Although previous reviews have been published,[42] a strength of our study is that we used reproducible systematic search techniques and reported quantitative comparisons to differentiate the compression or expansion of morbidity.

We chose to focus on evidence published from 2016, in order to avoid duplication of previous work and focus on contemporary evidence. A limitation of this approach is that we have omitted potentially useful studies published prior to 2016. However, this criterion has enabled a synthesis that prioritises the most recent evidence on trends in health expectancies. In addition to searching for published studies, we searched the Office for National Statistics website, the UK government body responsible for analysing data about the UK population. We did not search the equivalent national statistical bodies for every other high-income country as the volume of translation this would necessitate was unfeasible. However, this is not a major shortcoming: most of the non-UK studies used nationally representative samples, and none indicated unusual or contradictory trends that required verification from other data sources.

A key limitation of our synthesis is that while we were able to make some broad comparisons between studies, a more detailed comparison of trends was not possible. This was due to the high degree of heterogeneity between studies in: the age at which the health expectancy is estimated, trend periods (years and time frame), the measures used and the way trends were stratified. This not only prevented a more meaningful comparison, but also limited the extent to which we were able to explain differences in these trends between studies and countries. A further consideration here is the comparability of the overlapping and non-overlapping trend periods within and between countries in the studies included in this review (online supplemental figure 1). Different countries face national challenges at different times, while global challenges can differentially affect countries across the same time periods. Therefore, even similar trends across the same time periods may be due to different reasons.

Finally, the quality of this evidence was judged using criteria designed to assess threats to the validity of trends and the adequacy of the outcome measure.[12] For some studies, it was not possible to arrive at a clear judgement due to the absence of required methodological detail regarding survey response rates. This is an important limitation, but one that is easily addressed if future studies of health expectancy trends report the methodological detail required. Where a judgement on quality was possible, studies were generally of fair or good quality, signalling no major concerns to the validity of the

evidence synthesised. The rating of poor for a minority of studies largely reflected a change in the outcome measure over time,[26 29 31] which may reduce comparability between observed time points. To some extent, such changes are expected as panel and longitudinal studies evolve at each new point of data collection. Such reduced comparability may undermine the validity of the trends reported in these studies. However, this should be balanced against the judgements for the other quality assessment criteria, where methods were rated favourably and indicated minimal bias (see online supplemental materials). Overall, for studies where a clear judgement on quality was possible, we did not identify any studies that were particularly concerning in terms of quality and validity.

## CONCLUSIONS

In a number of high-income countries, changes in health expectancies over time have not kept pace with the growth in life expectancy. That is, people are living longer but disability and poor health are occupying an increasing proportion of later life. A compression of disability for women in Sweden signals some progress in achieving healthier longer lives. These findings have implications for health and care-related policy, and in particular ensuring people have timely access to care in later life. Further scrutiny of health expectancies in the wake of the COVID-19 pandemic is necessary given the high death rates in some countries and the long-term health consequences of the virus that are becoming apparent.

**Contributors** CJ planned the study, developed the protocol and co-wrote the paper; GFS developed the protocol, contributed to all stages of the review and co-wrote the paper; TPK developed the protocol, contributed to all stages of the review and co-wrote the paper; FB developed the protocol, developed and undertook the searches and co-wrote the paper; DC developed the protocol, oversaw all stages of the review and co-wrote the paper; BH developed the protocol and co-wrote the paper.

**Funding** National Institute for Health Research (NIHR) Policy Research Programme conducted through the NIHR Older People and Frailty Policy Research Unit, PR-PRU-1217-21502. The views expressed are those of the authors and not necessarily those of the NIHR or the Department of Health and Social Care.

**Competing interests** None declared.

**Patient consent for publication** Not required.

**Provenance and peer review** Not commissioned; externally peer reviewed.

**Data availability statement** All data relevant to the study are included in the article or uploaded as supplementary information. All data relevant to the study are included in the article or uploaded as supplementary information.

**ORCID iDs**
Gemma F Spiers http://orcid.org/0000-0003-2121-4529
Tafadzwa Patience Kunonga http://orcid.org/0000-0002-6193-1365
Barbara Hanratty http://orcid.org/0000-0002-3122-7190

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
