## [Reviewer comments · BMJ Open]

ARTICLE DETAILS

TITLE (PROVISIONAL)	Trends in health expectancies: a systematic review of international evidence
AUTHORS	Spiers, Gemma; Kunonga, Tafadzwa; Beyer, Fiona; Craig, Dawn; Hanratty, Barbara; Jagger, Carol

VERSION 1 – REVIEW

REVIEWER	Danny Dorling University of Oxford, England
REVIEW RETURNED	14-Oct-2020

GENERAL COMMENTS	I have attached my comments and also pasted them below. I have attached my comments as a file as well as the formatting of the table below will almost certainly be lost in the box for comments. I only have three comments – I like the paper overall and this it is very useful and represents a lot of hard work. My third comment explains why I ticked the No box on up to date references. 1) Why start the literature review in 2016; why not 2014, or 2017? The reason you give is “As this systematic review updated previous Reviews” but the two you mention (7 and 9) were published in 2016 and 2015 so presumably omitted some 2015 studies. Can you make this clear and also say if those reviews concluded the earlier studies (prior to 2015) were not very useful? 2) Just after you insert Table 3 you talk about disability-free life expectancy gains in the UK being slower than life expectancy gains; but life expectancy peaked in the UK in 2014 and has been lower every year since. It is possible that when a figure is published for 2019 it will be higher than 2014; but 2020 will be lower. See: Hiam, L., Dorling, D. and McKee, M. (2020) Things fall apart: The British Health Crisis 2010-2020, British Medical Bulletin, March 27th. https://academic.oup.com/bmb/article/133/1/4/5812717 Table 1 Life expectancy at birth for males and females, 2014 and 2018 Life expectancy at birth Change 2014–2018 2014 2018 Years Days of life Men Women Men Women Men Women Men Women England 79.51 83.23 79.55 83.20 0.04 -0.03 15 -11 Northern Ireland 78.61 82.38 78.84 82.44 0.23 0.06 84 22 Scotland 77.32 81.34 77.05 81.01 -0.27 -0.33 -99 -121 Wales 78.79 82.61 78.23 82.19 -0.56 -0.42 -205 -153 UK 79.25 82.99 79.24 82.93 -0.01 -0.06 -4 -22 Source: Authors’ calculations using ONS (2019) Single-year life tables, the UK: 1980–2018, released September 25, 2019, for the
--

	UK and all its separate countries: https://www.ons.gov.uk/peoplepopulationandcommunity/birthsdeaths 3) There is also evidence of life expectancy falling in the USA since 2014. Again this makes it hard to talk about how the gains of one kind of expectancy relate to another if there have not been gains. How All your USA papers are from 2016 (when data from 2014 was only just available. Was nothing published in the USA on this after 2016 but four papers were in 2016: 37 Cao B. Future healthy life expectancy among older adults in the US: a forecast based on cohort smoking and obesity history. Popul Health Metr 2016;14:23. 38 Crimmins EM, Zhang Y, Saito Y. Trends Over 4 Decades in Disability-Free Life Expectancy in the United States. Am J Public Health 2016;106:1287-93. 39 Freedman VA, Spillman BC. Active life expectancy in the older US population, 1982-2011 : differences between blacks and whites persisted. 2016. 40 Freedman VA, Wolf DA, Spillman BC. Disability-Free Life Expectancy Over 30 Years: A Growing Female Disadvantage in the US Population. Am J Public Health 2016;106:1079-85. Finally a general comment – we may well label things as disability when people are old but not when they are young. We tend not to count five years living with extreme anxiety as a twenty year old as years of healthy life lost; but given the rise in poor mental health and also of deaths from disease of despair in middle age – perhaps we should not label aspects of life that just come with aging as disabilities and we should label unnecessary suffering at younger ages as a disability more clearly in calculations such as these? I hope this paper is published at some point soon and that these comments are useful. - The reviewer provided a marked copy with additional comments. Please contact the publisher for full details.
--	---

REVIEWER	Deeg, Dorly VU University Medical Centre Amsterdam
REVIEW RETURNED	23-Dec-2020

GENERAL COMMENTS	This systematic review of trends in healthy and disability-free life expectancy (briefly, HLE) is timely, because many new trend-studies have been published even in the relatively short period that the authors cover, i.e., since 2016. It is good to step back and see what insights this – generally descriptive – literature had yielded. Unfortunately, I do not think that this review, in its present state, adds to this literature. I state some major concerns, as well as some minor issues. Major issues 1. In view of the generally descriptive nature of studies in this field, one would expect from a review that it goes beyond description and offers some analysis of the correspondence or contrasts between the various studies. Unfortunately, I see very little of that in this manuscript. For example, on p. 5, first sentence, the authors state that it is important to ‘identify and learn from countries whose populations are living healthier for longer’. It turns out that only few studies show greater or equal improvements in HLE than in life expectancy (LE), one of which has sufficient quality: the study on Swedish women. The overwhelming majority of studies show the
--

	opposite: smaller (or no) improvements in HLE than in LE. So if we should learn how to turn this trend around, we should have a closer look at the Swedish study. E.g., how do the Swedish authors explain their findings? Is the age range or the period covered different from the other studies? What does this tell us about how to conduct future studies? As is, the manuscript remains rather general and even repetitive in its Discussion. The authors could use the space much better by presenting a more thorough analysis. 2. The authors make an attempt to rate the quality of the studies included. In principle, this is an essential step in a systematic review. They established seven criteria (one of which is only applicable to longitudinal studies), from which they decide to use only four for a quality rating (p. 6). Why only four, and why not all seven, or six as applicable? Surprisingly, they then labeled as 'unclear' all studies for which they rated at least one of the four criteria as unclear. This is a rather strict rule, and application of this rule amounted to half of the studies being labeled as unclear (15 out of 30). This was such a large proportion that the authors decided not to prioritize evidence from studies with a higher quality rating in their review of the findings. I think this is a strange way of dealing with the available information. Instead, the rules for applying the quality criteria should be such that it enables a quality rating for most of the studies included. 3. Another issue concerning the label 'unclear' is that it is good practice for authors doing a systematic review, to go back to the authors of an article included in the review and ask for clarification. Even with the current strict rule for 'unclear', this should have substantially decreased the studies labeled as unclear. In fact, I consider it unfair to the authors of the original studies to refrain from asking for feedback, because for some studies listed in Suppl Table 2, I know that they have clearly reported the criteria that the authors of this Table have labeled as unclear. 4. In the list of quality criteria, I do miss two that are essential to interpret the results from the studies included. First, is the trend based on only two time points, or more? In order to really know a trend, I consider a study using only two time points as insufficient. In the real world, estimates of health fluctuate over the years, and two time points do not reveal fluctuations, but can erroneously lead to a conclusion on a trend in a certain direction. Second, what was the non-response in repeated cross-sectional studies? Now, only attrition in longitudinal studies is rated as a criterion. 5. An issue that the authors ignore is what period the trend study covers. Trends may differ across periods. In Table 3, the first time points range from 1970 to 2009/11, and the last time points, from 1999/2000 to 2017. Thus, the studies included consider non-overlapping period, which should be taken into account in the review of the finding. 6. In the review criteria, it is not clear what aspects of 'health' are included. I note that from some studies, data on cognitive functioning are reported (Jagger 2016, Grasset 2019, see Suppl Table 1; also Deeg 2018, see Table 3). This is not stated in the Methods, nor in Table 3 (with one exception). However, it is known that trends in LE free from cognitive impairment and dementia follow a different pattern than trend in HLE. Mixing these trends obscures potential mechanisms. Moreover, there are several other studies on LE free from cognitive impairment or dementia that I know of that were published since 2016. Thus, if the authors wish to consider trends in cognitively healthy LE, they should be more systematic in doing so, or else leave these for another review.
--	---

	Other comments;  1. The last sentence of the Abstract on Covid19 is rather speculative, and does not state an expected direction of the likely impact. This could be better replaced by a more informative concluding sentence. 2. In the first paragraph of the Introduction, the authors seem to suggest that the demographic shift towards a higher proportion of older people in the population implies the need for more care. This statement assumes that older people are (and will remain) in poor health. However, this is exactly the research question for a study on trends in HLE. 3. It is not clear why the authors include regular updates of trends from the UK Office of National Statistics only. Many other national statistical offices in Europe publish regular updates, and so does Eurostat. Has this manuscript perhaps been based on a report to a UK agency, extended with studies elsewhere? As is, it provides an unbalanced overview. 4. In Suppl Table 1, the column 'Reports evidence about factors associated with trends' lists only 'No', except for one case (a projection study). This column does not seem very useful. 5. On page 8, the paragraph on Table 3 should be placed before the previous paragraph, which summarized the findings across studies. 6. Table 3 and the larger tables in the Supplement are hard to read. Please use the Word command 'Repeat Header Rows' for better readability. 7. In Table 3, the column 'Trend period' often lists two years of measurement, presumably the first and the last year of the period considered. However, for some studies more years of measurement are listed (up to five: Crimmins 2016). This suggests that all other studies are based on two measurements only, but to my knowledge several studies listed as such have more than two years of measurement. 8. The last paragraph of the Results is about countries with at least two studies showing contrasting findings. (The last sentence is about 'the two studies from the Netherlands', but you included three for this country). Again, the periods covered and/or the ages from which HLE is calculated differ across the studies, not to speak of the health measures. Before you conclude that they show contrasting trends, you need to consider period, age, and health measure, instead of dismissing these studies as 'limiting any inference'. 9. In the discussion of limitations, the reduced comparability is stated to be balanced by 'other aspects' (p. 12). Which are these? In conclusion, I consider this (more or less) systematic review as important and timely, but the authors should do a much better job in reviewing the findings from the studies included.
--	--

REVIEWER	Minsu, Ock University of Ulsan College of Medicine
REVIEW RETURNED	03-Jan-2021

GENERAL COMMENTS	Review comments: Authors conducted a systematic review to update and summarise evidence on trends in healthy and disability-free life expectancy, in OECD high-income countries. I appreciate authors' hard work, but authors need to find the implications of a systematic review for healthy life expectancy.
---

	First of all, the conceptual definition of healthy life expectancy needs to be clarified. I know disability-free life expectancy as an example of healthy life expectancy. What is the healthy life expectancy the authors are talking about? Quality-adjusted life expectancy, disease-free life expectancy and life expectancy taking into account self-rated health are also examples of healthy life expectancy. The authors seem to need to clarify the concept of healthy life. The indicators of healthy life expectancy used by the previous literature are different, but I am not sure what it means to simply present the results. It seems that it will be much more meaningful to highlight the differences in the specific methods used to calculate the healthy life expectancy rather than the presented results of healthy life expectancy. Thank author for the opportunity to review the interesting manuscript.
--	--

VERSION 1 – AUTHOR RESPONSE

Reviewer 1	
Why start the literature review in 2016; why not 2014, or 2017? The reason you give is “As this systematic review updated previous Reviews” but the two you mention (7 and 9) were published in 2016 and 2015 so presumably omitted some 2015 studies. Can you make this clear and also say if those reviews concluded the earlier studies (prior to 2015) were not very useful?	We apologise as one of the references cited is incorrect (Jagger et al, 2016) and should be Kingston et al. (2017) – this publication included a search for studies dated from January 1st 2009 to December 31 2016 as part of a Lancet Research In Context (see doi: http://dx.doi.org/10.1016/). Searches were carried out to follow on from this. We have corrected this reference in the paper. The previous reviews we refer to did not make any conclusions about the usefulness of the studies/data, so we are unable to comment on this. One (Jagger, 2015) is a review of data rather than published studies, whilst the second (Kingston, 2017) is a review of published studies as

	part of a Lancet Research in Context. We accept that by omitting studies prior to 2016, we may have overlooked potentially useful evidence, but we intended to capture the most up to studies. We have amended our statement about this in the discussion to acknowledge this limitation: We chose to focus on evidence published from 2016, in order to avoid duplication of previous work and focus on contemporary evidence. A limitation of this approach is that we have omitted potentially useful studies published prior to 2016. However, this criterion has enabled a synthesis that prioritises the most recent evidence on trends in healthy and disability-free life expectancy.
Just after you insert Table 3 you talk about disability-free life expectancy gains in the UK being slower than life expectancy gains; but life expectancy peaked in the UK in 2014 and has been lower every year since. It is possible that when a figure is published for 2019 it will be higher than 2014; but 2020 will be lower. See: Hiam, L., Dorling, D. and McKee, M. (2020) Things fall apart: The British Health Crisis 2010-2020, British Medical Bulletin, March 27th. https://academic.oup.com/bmb/article/133/1/4/5812717	Yes this is possible but, certainly for the UK, it has also been the case that disability-free life expectancy has fallen too as shown in a very recent paper: Claire E. Welsh, Fiona E. Matthews, Carol Jagger, Trends in life expectancy and healthy life years at birth and age 65 in the UK, 2008–2016, and other countries of the EU28: An observational cross-sectional

	study, The Lancet Regional Health - Europe, Volume 2, 2021, 100023, https://doi.org/10.1016/j.lanepe.2020.100023. We have not added anything further in the paper because of space constraints.
There is also evidence of life expectancy falling in the USA since 2014. Again this makes it hard to talk about how the gains of one kind of expectancy relate to another if there have not been gains. How All your USA papers are from 2016 (when data from 2014 was only just available. Was nothing published in the USA on this after 2016 but four papers were in 2016: 37 Cao B. Future healthy life expectancy among older adults in the US: a forecast based on cohort smoking and obesity history. Popul Health Metr 2016;14:23. 38 Crimmins EM, Zhang Y, Saito Y. Trends Over 4 Decades in Disability-Free Life Expectancy in the United States. Am J Public Health 2016;106:1287-93. 39 Freedman VA, Spillman BC. Active life expectancy in the older US population, 1982-2011 : differences between blacks and whites persisted. 2016. 40 Freedman VA, Wolf DA, Spillman BC. Disability-Free Life Expectancy Over 30 Years: A Growing Female Disadvantage in the US Population. Am J Public Health 2016;106:1079-85.	There are a number of reasons why there are no more recent analyses of trends in disability-free life expectancy. Firstly in some countries there is a lag in the production of national mortality/life expectancy data – this is particularly true for the UK so that Eurostat values for HLY are typically at least two years behind the current date. Secondly for disability-free life expectancy the disability prevalence comes from surveys. In many countries these will not be annual surveys (though they are for the national UK estimates). However we take the reviewers comments that there might not have been gains and have amended this to ‘changes’ where necessary.
Finally a general comment – we may well label things as disability when people are old but not when they are young. We tend not to count five years living with extreme anxiety as a twenty year old as years of healthy life lost;	Disability is a very broad term and one that has been challenged. The questions in surveys of late life individuals tend to focus on specific activities of daily living but

but given the rise in poor mental health and also of deaths from disease of despair in middle age – perhaps we should not label aspects of life that just come with aging as disabilities and we should label unnecessary suffering at younger ages as a disability more clearly in calculations such as these?	the surveys of all adults would use more general activity limitation questions (such as the one underlying the EU Healthy Life Years or the UK Limiting Long-standing illness question). There has been validation of all of these measures for different age groups. We have not added anything further to the manuscript regarding this point, due to space constraints.
Reviewer 2	
This systematic review of trends in healthy and disability-free life expectancy (briefly, HLE) is timely, because many new trend-studies have been published even in the relatively short period that the authors cover, i.e., since 2016. It is good to step back and see what insights this – generally descriptive – literature had yielded. Unfortunately, I do not think that this review, in its present state, adds to this literature. I state some major concerns, as well as some minor issues.	Thank you for your thoughtful and constructive comments. Where possible, we have made revisions in response to these and summarise these below.
In view of the generally descriptive nature of studies in this field, one would expect from a review that it goes beyond description and offers some analysis of the correspondence or contrasts between the various studies. Unfortunately, I see very little of that in this manuscript. For example, on p. 5, first sentence, the authors state that it is important to ‘identify and learn from countries whose populations are living healthier for longer’. It turns out that only few studies show greater or equal improvements in HLE than in life expectancy (LE), one of which has sufficient quality: the study on Swedish women. The overwhelming majority of studies show the	We agree that some degree of comparison between studies could offer a more analytical perspective, but as we highlight in the discussion section, we were unable to make these comparisons due to the differences between studies in the age at which the health expectancy is estimated, trend periods (years and time frame), the measures used, and the way trends were stratified. This is a key limitation of this work, which

opposite: smaller (or no) improvements in HLE than in LE. So if we should learn how to turn this trend around, we should have a closer look at the Swedish study. E.g., how do the Swedish authors explain their findings? Is the age range or the period covered different from the other studies? What does this tell us about how to conduct future studies? As is, the manuscript remains rather general and even repetitive in its Discussion. The authors could use the space much better by presenting a more thorough analysis.

does inevitably lead to a more descriptive, rather than analytical, review, but one which we fully acknowledge in the discussion.

With regards to the Sundberg study (compression of disability for women in Sweden), we have added some further context to the finding in the discussion section (yellow highlight):

One clear exception was Sweden, where gains in women's disability-free life expectancy were greater than gains in life expectancy over a period of almost 20 years. This was a finding evidenced from two different datasets within Sundberg and colleagues' study (SWEOLD: 1992-2011; SHARE: 2004-2011). This finding contrasts with an earlier review, which found evidence of a compression of disability for men, but not women, in Sweden.[9] This has now been reversed with an expansion of disability for men.[35] Sundberg and colleagues attribute the compression of disability for women to improved health for women but not men, whilst men were living longer due to falling deaths from cardiovascular disease.

The authors make an attempt to rate the quality of the studies included. In principle, this is an essential step in a systematic review. They established seven criteria (one of which is only applicable to longitudinal studies), from which they decide to use only four for a quality rating (p. 6). Why only four, and why not all seven, or six as applicable? Surprisingly, they then labeled as 'unclear' all studies for which they rated at least one of the four criteria as unclear. This is a rather strict rule, and application of this rule amounted to half of the studies being labeled as unclear (15 out of 30). This was such a large proportion that the authors decided not to prioritize evidence from studies with a higher quality rating in their review of the findings. I think this is a strange way of dealing with the available information. Instead, the rules for applying the quality criteria should be such that it enables a quality rating for most of the studies included.

Another issue concerning the label 'unclear' is that it is good practice for authors doing a systematic review, to go back to the authors of an article included in the review and ask for clarification. Even with the current strict rule for 'unclear', this should have substantially decreased the studies labeled as unclear. In fact, I consider it unfair to the authors of the original studies to refrain from asking for feedback, because for some studies listed in Suppl Table 2, I know that they have clearly reported the

As we describe in the methods section, studies did not typically report three of the seven criteria: *proportion lost to follow up*, *proportion of proxy interviews* and *proportion of missing data*. Since most of the trends are based on cross-sectional studies and national mortality data, loss to follow-up is irrelevant, and the other two criteria are rarely, if ever, reported, but are unlikely to be large. This is why we chose to base the quality assessment on the criteria for which there was detail in publications and/or technical annexes.

However, we appreciate your suggestion of including two further criteria, which we have added to the quality assessment. We have also attempted to contact authors for further detail. To explain the addition of these criteria to an existing approach to assessing bias in studies of trends, we have noted the following in the methods:

Following expert advice, we also judged two further criteria: whether the trend analysis used more than two time points, and the % of non-response in repeated cross-section surveys.

Unfortunately, this still resulted in there being insufficient information for the criteria *proportion of proxy interviews* and *proportion of missing data* for almost half of studies. Given these limitations, we have assessed the studies using the criteria we think are most important for ascertaining bias: comparability of interview methods, quality of the outcome measure, whether more than two time

criteria that the authors of this Table have labeled as unclear. In the list of quality criteria, I do miss two that are essential to interpret the results from the studies included. First, is the trend based on only two time points, or more? In order to really know a trend, I consider a study using only two time points as insufficient. In the real world, estimates of health fluctuate over the years, and two time points do not reveal fluctuations, but can erroneously lead to a conclusion on a trend in a certain direction. Second, what was the non-response in repeated cross-sectional studies? Now, only attrition in longitudinal studies is rated as a criterion.	points were used (as requested), and the response rate in repeated cross sectional studies or loss to follow up for longitudinal studies. Where authors were able to provide the detail about the proportion of proxy respondents and missing data, or this was provided in the publication(s), we have included this in the table for transparency. We also highlighted the limitation of the quality assessment in the discussion as well as signalling the need for more detail reported in publications to allow an assessment of quality. It also highlights the difficulty of assessing quality in studies of trends in indicators such as DFLE.
An issue that the authors ignore is what period the trend study covers. Trends may differ across periods. In Table 3, the first time points range from 1970 to 2009/11, and the last time points, from 1999/2000 to 2017. Thus, the studies included consider non-overlapping period, which should be taken into account in the review of the finding.	Different countries face national challenges at different times and even global challenges can differentially affect countries across the same time periods. Therefore even similar trends across the same time periods may be due to different reasons. We have added this point to the discussion and also a figure which better shows the periods covered and the compression or expansion:

	A further consideration here is the comparability of the overlapping and non-overlapping trend periods within and between countries in the studies included in this review (supplementary materials figure 2). Different countries face national challenges at different times, whilst global challenges can differentially affect countries across the same time periods. Therefore even similar trends across the same time periods may be due to different reasons.
In the review criteria, it is not clear what aspects of ‘health’ are included. I note that from some studies, data on cognitive functioning are reported (Jagger 2016, Grasset 2019, see Suppl Table 1; also Deeg 2018, see Table 3). This is not stated in the Methods, nor in Table 3 (with one exception). However, it is known that trends in LE free from cognitive impairment and dementia follow a different pattern than trend in HLE. Mixing these trends obscures potential mechanisms. Moreover, there are several other studies on LE free from cognitive impairment or dementia that I know of that were published since 2016. Thus, if the authors wish to consider trends in cognitively healthy LE, they should be more systematic in doing so, or else leave these for another review.	We have amended our terminology throughout the paper to refer to ‘health expectancies’, which we use to encompass healthy and disability free life expectancy. We have referred to the specific type of health expectancy (e.g. disability free life expectancy, healthy life expectancy), where necessary. We have also added further detail to what we considered eligible health expectancies in the methods: Studies were included if they examined trends (i.e. more than one time point) in health expectancies. Eligible health expectancies were general healthy life expectancy, disability free life expectancy, active life expectancy, health-related quality adjusted life expectancy, and health adjusted life expectancy. We did not include healthy cognitive life expectancy, dementia free life

	expectancy, or life expectancy with diseases. To ensure consistency we have removed studies and outcomes reporting LE free of cognitive impairment/dementia, and LE with diseases. These changes have been made to the main text, table 3 and supplementary materials tables.
. The last sentence of the Abstract on Covid19 is rather speculative, and does not state an expected direction of the likely impact. This could be better replaced by a more informative concluding sentence.	We have amended this sentence to: The recent COVID-19 pandemic may adversely impact health expectancies in the longer term. We have also added a sentence to the discussion section where we refer to COVID-19 to report recent analysis showing the impact of COVID on life expectancy in the UK: A recent analysis of UK mortality data in 2020, for example, indicates that COVID has reduced life expectancy by around a year.[43]
In the first paragraph of the Introduction, the authors seem to suggest that the demographic shift towards a higher proportion of older people in the population implies the need for more care. This statement assumes that older people are (and will remain) in poor health. However, this is exactly the research question for a study on trends in HLE.	We intended this sentence to point towards the potential implications of increased population ageing for health and care. We have reworded this sentence to reflect this: This demographic shift will require societies to adapt. If longer lives are spent in poor health, governments face the challenge of providing accessible,

	high quality and sustainable long-term care.[2-4]
It is not clear why the authors include regular updates of trends from the UK Office of National Statistics only. Many other national statistical offices in Europe publish regular updates, and so does Eurostat. Has this manuscript perhaps been based on a report to a UK agency, extended with studies elsewhere? As is, it provides an unbalanced overview.	We agree that this is a limitation, which we acknowledged in the discussion section. We also explain in the discussion that we did not search other statistical offices as the volume of translation necessary was unfeasible for the project.
In Suppl Table 1, the column 'Reports evidence about factors associated with trends' lists only 'No', except for one case (a projection study). This column does not seem very useful.	We have removed this column from Table 1.
On page 8, the paragraph on Table 3 should be placed before the previous paragraph, which summarized the findings across studies.	The paragraph on page 8, which refers to table 3, is the start of this section of the findings that describe the difference between each health expectancy and total life expectancy (under the section header Is there evidence of an expansion or compression of disability?). We feel it is more appropriate here than the previous paragraph, which gives a summary overview of all studies.
Table 3 and the larger tables in the Supplement are hard to read. Please use the Word command 'Repeat Header Rows' for better readability.	We have amended all supplementary materials tables that extend to two or more pages so that header rows are repeated.
In Table 3, the column 'Trend period' often lists two years of measurement, presumably the first and the last year of the period considered. However, for some studies more years of measurement are listed (up to five: Crimmins 2016). This suggests that all other studies are based on two	We apologise for this oversight, and we have amended this table to include all years listed within the trend period.

measurements only, but to my knowledge several studies listed as such have more than two years of measurement.	
The last paragraph of the Results is about countries with at least two studies showing contrasting findings. (The last sentence is about ‘the two studies from the Netherlands’, but you included three for this country). Again, the periods covered and/or the ages from which HLE is calculated differ across the studies, not to speak of the health measures. Before you conclude that they show contrasting trends, you need to consider period, age, and health measure, instead of dismissing these studies as ‘limiting any inference’.	We have amended this section to add further detail. Regarding the contrast in findings for the two studies from Korea, we have added the following (highlighted in yellow): Evidence from Korea and the Netherlands was less certain. Contrasting findings between studies gave an unclear picture about whether these countries had experienced a compression or expansion of disability. For example, in one study from Korea, gains in quality adjusted life expectancy at birth were greater than gains in life expectancy.[32] Another study, using the same data but a different measure (healthy life expectancy), reported that gains at birth still lagged behind those for life expectancy, although gains in healthy life expectancy at age 65 and 85 were slightly higher than gains in life expectancy over the study period.[33] This contrast may reflect the difference in measures; Jo and colleagues suggest that quality adjusted life expectancy may be overestimated using the EQ-5D-3L, which forms the basis of this measure. Regarding the contrasting findings between the studies for the Netherlands, we have amended this section:

	Three studies from the Netherlands also offered inconsistent findings: this may reflect differences in the trend periods, age at which expectancies were estimated and the measures used. For example, Reus-Pons and colleagues demonstrated gains in healthy life expectancy were smaller than gains in total life expectancy at age 50, over a ten year period (2001-2011). Deeg and colleagues examined physical HLE over a 23 year period (1993-2016): a decline was observed, whilst total life expectancy increased. Gheorghe and colleagues estimated quality adjusted life expectancy at age 25 and 65, and stratified by educational attainment. These trends indicated an expansion of poor health at age 25 and 65 for men at all levels of educational attainment, but a compression of poor health for women at age 25 (all levels of education) and 65 (high and low education).
In the discussion of limitations, the reduced comparability is stated to be balanced by 'other aspects' (p. 12). Which are these?	We have amended this section to make clear we are referring to the other criteria in the quality assessment: Such reduced comparability may undermine the validity of the trends reported in these studies. However, this should be balanced against the judgements for the other quality assessment criteria, where methods were rated favourably and indicated minimal bias.
Reviewer 3	
Authors conducted a systematic review to update and summarise evidence on trends in healthy and	The value of this review is that we have presented an up to date picture of trends in health expectancies using systematic

disability-free life expectancy, in OECD high-income countries. I appreciate authors' hard work, but authors need to find the implications of a systematic review for healthy life expectancy.	reproducible searches, including an assessment of whether trends in healthy expectancies are keeping pace with total life expectancy and an assessment of the quality of the studies. Although challenging, this is the first attempt to assess the quality of studies reporting trends in health expectancies in a systematic review. Overall, given the evolving evidence base, we believe this systematic review a timely and useful contribution.
First of all, the conceptual definition of healthy life expectancy needs to be clarified. I know disability-free life expectancy as an example of healthy life expectancy. What is the healthy life expectancy the authors are talking about? Quality-adjusted life expectancy, disease-free life expectancy and life expectancy taking into account self-rated health are also examples of healthy life expectancy. The authors seem to need to clarify the concept of healthy life.	We have amended our terminology throughout the paper to refer to 'health expectancies', which we use to encompass healthy and disability free life expectancy. We have referred to the specific type of health expectancy (e.g. disability free life expectancy, healthy life expectancy), where necessary. We have also added further detail to defined what we considered eligible health expectancies in the methods: Studies were included if they examined trends (i.e. more than one time point) in health expectancies. Eligible health expectancies were general healthy life expectancy, disability free life expectancy, active life expectancy, health-related quality adjusted life expectancy, and health adjusted life expectancy. We did not include healthy cognitive life expectancy, dementia free life expectancy, or life expectancy with diseases.
The indicators of healthy life expectancy used by the previous literature are different, but I am not	All but one study calculated life expectancy using the Sullivan method, so there is almost no between study

sure what it means to simply present the results. It seems that it will be much more meaningful to highlight the differences in the specific methods used to calculate the healthy life expectancy rather than the presented results of healthy life expectancy.	differences to highlight in this respect. We have added a column to supplementary materials table 1 to include the method used to calculate the expectancy. In terms the measurement of health, this is too diverse between studies to draw out meaningful differences. Even where studies use activities of daily living, studies differ in terms of what activities are included in the question.
---	--

VERSION 2 – REVIEW

REVIEWER	Deeg, Dorly VU University Medical Centre Amsterdam
REVIEW RETURNED	24-Mar-2021

GENERAL COMMENTS	The authors have responded very well to my comments. Regardless, in the current version a few (minor) points need to be addressed.  1. The concluding sentence in the abstract is too speculative. The authors have added a reference on loss of one year life expectancy for the UK, but a shorter life expectancy due to infectious diseases may well mean the same or even better health expectancy (see Robine & Michel, J Gerontol Med Sci 2004). 2. In the Summary box, the last bullet still states that the quality of half of the studies could not be assessed. This is now 10 out of 28. 3. The quality criterion 'loss to follow-up' for longitudinal seems to include loss due to mortality. This definition of loss to follow-up ignores that mortality also happens in the population to which the study sample is supposed to generalize. In all tables where 'change' is reported, 'change' should be defined correctly in a footnote. For example in Suppl tables 3 through 7, 'change' is specified as 'between first and last time point'. However, the differences in this column suggest that it is instead: between last and first time point.
--

VERSION 2 – AUTHOR RESPONSE

Response document

Comment	Response
The concluding sentence in the abstract is too speculative. The authors have added a reference on loss of one year life expectancy for the UK, but a shorter life expectancy due to	We have amended the sentence in the abstract to:

infectious diseases may well mean the same or even better health expectancy (see Robine & Michel, J Gerontol Med Sci 2004).	The recent COVID-19 pandemic may also impact health expectancies in the longer term. We have also added a sentence to the main text where we refer to the impact of COVID: Finally, given the high death rates in some countries as a result of the COVID-19 pandemic, and the likely long-term health consequences of the virus that are becoming apparent, further scrutiny of future trends in health expectancies is needed. A recent analysis of UK mortality data in 2020, for example, indicates that COVID has reduced average life expectancy by around a year.[41] The ways in which this may impact health expectancies has yet to be determined.
In the Summary box, the last bullet still states that the quality of half of the studies could not be assessed. This is now 10 out of 28.	Thank for you highlighting this, we have amended to: Due to the absence of methodological detail reported, it was not possible to give a clear judgement of study quality and bias for 10 of 28 studies included in the review.
The quality criterion 'loss to follow-up' for longitudinal seems to include loss due to mortality. This definition of loss to follow-up ignores that mortality also happens in the population to which the study sample is supposed to generalize.	We did not define loss to follow up as meaning with or without mortality. We agree that mortality data do contribute in multi-state life tables. However only one study (Cao, 2016) used MSLT but used external data (and assumptions) to calculate mortality. In addition this study was not included in HE change as it estimated forecasted HEs. All other studies used repeated cross sectional analyses, so the figures reported were response rates not loss to follow up. However, four studies analysed used repeated cross-sections from longitudinal data. In this case including mortality as loss to follow-up is potentially important as a high loss through mortality would make subsequent waves unrepresentative of the total population at that time point, unless samples are refreshed across the whole age range.

In all tables where 'change' is reported, 'change' should be defined correctly in a footnote. For example in Suppl tables 3 through 7, 'change' is specified as 'between first and last time point'. However, the differences in this column suggest that it is instead: between last and first time point.

The change over time is from the first time point to last time point. The change is calculated as the first time point subtracted from the last time point.

We have added this statement within the supplementary materials prior to Table 3a:

For tables 3a to 7b, the change in expectancy from the first to the last time point represents the final estimated expectancy minus the first estimated expectancy.